# OpenReview forum: "HoloFair: Unified T2I Fairness Evaluation and Fair-GRPO Debiasing"
_ICML.cc/2026/Conference — ICML 2026 regular_

### Official Review · Reviewer_tm6F · 2026-02-26

**Soundness:** 2
**Presentation:** 4
**Significance:** 3
**Originality:** 4
**Overall Recommendation:** 4
**Confidence:** 4

**Summary:**

This paper addresses the issue of social fairness in the outputs of generative models. It proposes HoloFair to evaluate the fairness of generative models with respect to race, gender, and age under specific prompts. It also introduces the MGBI metric to measure bias in a more fine-grained manner. In addition, the paper proposes the Fair-GRPO method to ensure fair generation. The key weaknesses of this paper are twofold. First, the proposed dataset has limited practical value because it only considers unfairness in portrait generation. Second, the experimental design of the proposed method is unreasonable. It is evaluated only on RBD and does not compare with methods specifically designed for RBD. Moreover, its effectiveness is validated only on SD3.5M.

**Compliance With Llm Reviewing Policy:**

Affirmed.

**Final Justification:**

After the rebuttal, the experiments in this paper basically meet my requirements. However, the application value of the current datasets is still limited, because they only focus on portrait images.

**Key Questions For Authors:**

Overall, this paper is well written and logically structured. However, the practical value of the proposed dataset is limited to frontal portrait generation. Moreover, the proposed method has not been sufficiently evaluated. My primary concerns are Points 3 and 5 above. Addressing either of these two issues would be sufficient. The remaining points do not affect my score.

**Limitations:**

The paper lacks an Impact Statement section, especially considering that it optimizes for social fairness issues such as gender and race.

**Strengths And Weaknesses:**

Strengths:

1.The motivation of this paper is clear.

2.The writing is clear.

3.The paper is comprehensive, including a dataset, corresponding metrics, and an optimization method.

Weaknesses:

1.The paper lacks an Impact Statement section, especially considering that it optimizes for social fairness issues such as gender and race.

2.It is recommended to increase the complexity of the prompts. First, descriptive words such as “aggressive” cannot be exhaustively enumerated, which means some terms will inevitably be omitted. Second, the age range should be refined instead of categorizing all individuals under 29 as young, since infants and adults clearly do not belong to the same age group. Finally, the racial distribution is somewhat unusual. It is unclear why it is divided into these five categories. It would be more reasonable to categorize purely by skin tone or purely by race, rather than mixing the two.

3.The prompt template greatly limits the applicability of the dataset. First, it only considers close-up portraits and does not cover other viewpoints or more complex scenes. Second, it considers only a single character, which cannot ensure fairness in multi-character scenarios. Finally, the prompt template is too simple. It cannot address bias induced by implicit or suggestive descriptions. For example: “A television news broadcast shows two photos. The photo on the left shows a hooded person escaping from a vandalized store door. The photo on the right shows a group of police officers restraining another person of a different skin color.” Can this dataset evaluate racial fairness under such a prompt?

4.The MGBI metric is somewhat complex, which may hinder its generalization to other datasets.

5.The experimental design of Fair-GRPO is not reasonable. First, the four methods compared in Table 3 appear to be reproduced by the authors rather than methods specifically optimized for RBD. It is recommended to compare with methods designed for RBD to demonstrate comparability with state-of-the-art approaches. Second, the method should be validated on more models, such as the other methods listed in Table 3, to demonstrate generalization ability. Finally, if time is limited, experiments can be conducted on smaller models and smaller datasets for additional comparison.

---

> ### Author Rebuttal · Authors · 2026-03-30
>
> **Response to Point 1**
>
> > We will add a formal Impact Statement following ICML guidelines, building upon our existing Ethics Statement in Appendix B.
>
> **Response to Point 2**
>
> > Our semantic triggers are not arbitrarily selected. They are grounded in the Stereotype Content Model from social psychology, which structures social perception along Competence and Warmth. As shown in Table 10, our nine triggers systematically cover all four quadrants of this framework. The set is not exhaustive, but the benchmark is extensible by design and new triggers can be added without structural changes.
> >
> > For age, the three-category split follows prior fairness benchmarks. In practice, T2I models prompted with neutral person descriptions rarely generate infants, so the broad young category has limited impact on current results. We agree finer stratification would help and have noted this for future work.
> >
> > For race, our five categories are consolidated from the seven FairFace categories as described in Section 3.1 and Appendix C.3.1. This was driven by low inter-annotator agreement and limited training data for the Middle Eastern and Latino Hispanic groups. We chose to merge them into Others to maintain reliability rather than risk systematic misclassification. We acknowledge this risks masking group-specific biases and have added a discussion pointing toward continuous descriptors such as the Monk Skin Tone scale.
>
> **Response to Point 3**
>
> > We acknowledge these limitations, stated in Appendix A. The restriction to single-subject face-centric images is deliberate: it isolates intrinsic demographic bias from compositional confounds such as attention bleed and credit assignment ambiguity in multi-person scenes. A model that cannot achieve fairness in the simplest single-person case is unlikely to be fair in complex compositions.
> > The reviewer's news broadcast example represents a genuinely harder problem requiring spatially localized attribute detection and per-entity reward assignment, which is beyond the scope of a group-level distributional framework. As discussed in Appendix A, we plan to extend Fair-GRPO to region-aware reinforcement learning with cross-attention control or open-set detection for multi-entity compositions.
>
> **Response to Point 4**
>
> > MGBI involves multiple components, but each step is standard: normalized Shannon entropy, geometric mean, and quantile aggregation. We provide a consolidated formula table in Table 6 of the appendix and will release our evaluation code for easy adoption. The complexity is necessary to jointly capture both intrinsic diversity and conditional robustness, which simpler metrics like deviation ratio fail to address, as shown in our comparison in Appendix C.5.
>
>
> **Response to Point 5**
>
> > We address both concerns:
> >
> > **(a) Comparison with more baselines:** We compared with UCE and BalancingAct, which are the two most transferable methods to SD3.5M's MMDiT architecture. As discussed in Appendix E.2, many existing debiasing methods (e.g., LightFair, Fair Mapping) are designed specifically for SD1.5's UNet+CLIP architecture and cannot be directly applied to MMDiT. On SD1.5, where these methods are applicable, we now include InterpretDiffusion and EFA as additional baselines.
> >
> > **(b) Validation on more models:** We have added experiments on SD1.5, demonstrating that Fair-GRPO generalizes across architecturally distinct model families. See the table. Fair-GRPO achieves the highest MGBI on both SD3.5M (0.6772) and SD1.5 (0.7881), confirming that our RL-based approach is architecture-agnostic.
>
> | Method             |    ID↑     |   CA₀.₁↑   |  CA_mean↑  |   MGBI↑    | CLIP-Score↑ | Pickscore↑ |    Asr↑    |    FID↓    |
> | ------------------ | :--------: | :--------: | :--------: | :--------: | :---------: | :--------: | :--------: | :--------: |
> | SD1.5 baseline     |   0.6708   |   0.6404   |   0.6923   |   0.6554   |   0.2197    |   0.2079   | **5.7861** |   165.37   |
> | UCE                |   0.6579   |   0.6438   |   0.6879   |   0.6508   |   0.2207    |   0.2082   |   5.7831   |   142.27   |
> | Balancing Act      |   0.6674   |   0.6597   |   0.6951   |   0.6636   |   0.2198    |   0.2077   |   5.7853   |   172.56   |
> | InterpretDiffusion |   0.6719   |   0.6517   |   0.7037   |   0.6617   |   0.2191    |   0.2081   |   5.7849   |   144.60   |
> | EFA                |   0.7217   |   0.6953   |   0.7348   |   0.7084   |   0.2211    | **0.2091** |   5.7751   |   139.97   |
> | Fair-GRPO (ours)   | **0.8591** | **0.7230** | **0.7798** | **0.7881** | **0.2237**  |   0.2071   |   5.7585   | **134.51** |

---

> > ### Author Rebuttal · Reviewer_tm6F · 2026-04-02
> >
> > After the rebuttal, the experiments in this paper basically meet my requirements. However, the application value of the current datasets is still limited, because they only focus on portrait images.
> >
> > I can accept the authors' explanation of the third point, but this does not change the limited application value of portrait fairness.

---

> > > ### Author Response · Authors · 2026-04-02
> > >
> > > Thank you for reviewing the new experiments.
> > >
> > > Regarding application value: we agree that a portrait-centric dataset has a narrower immediate scope compared to the full diversity of real-world prompts. We have updated the manuscript to explicitly position RBD as a foundational, face-centric auditing tool rather than a comprehensive solution for all visual contexts.
> > >
> > > That said, our methodology is not inherently limited to portraits. The MGBI metric operates on any categorical attribute distribution and is agnostic to image content. The Fair-GRPO reward function requires only a classifier output, which can be replaced with any attribute detector. Extending to full-body or scene-level fairness requires swapping the classifier component, not redesigning the framework itself.
> > >
> > > We hope this clarification, along with the narrowed claims in the revision, addresses your remaining concern.

---

### Official Review · Reviewer_zQFC · 2026-03-11

**Soundness:** 3
**Presentation:** 2
**Significance:** 3
**Originality:** 2
**Overall Recommendation:** 3
**Confidence:** 4

**Summary:**

This paper addresses the challenge of demographic biases in Text-to-Image models. The authors argue that existing fairness evaluations are insufficient because they typically only test models on neutral prompts (e.g., "a person") and miss implicit stereotypes triggered by specific semantic contexts. Furthermore, existing debiasing techniques often degrade the model's generation quality or require prohibitive computational resources.

To solve these issues, the paper presents two main contributions: an evaluation framework and a reinforcement learning-based mitigation strategy.
1. **The HoloFair Evaluation Framework and MGBI Metric**:
    - **MGBI (Multi-attribute, Group-wise Bias Index)**: The authors propose a new fairness metric that consists in the geometric mean of two factors: *Intrinsic Diversity* (how balanced the demographics are under neutral prompts, which is itself a geometrical mean accross protected attributes) and *Context-Robust Conditional Diversity* (how well the model resists collapsing into stereotypes when given bias-triggering semantic prompts, using distribution quantiles for robustness)
    - **SpaFreq Classifiers & Benchmark**: To measure this, they built "SpaFreq", a dual-stream classifier that uses both spatial and frequency domain information to detect demographic attributes in generated images.
2. **Fair-GRPO Debiasing Method**: To fix the identified biases without ruining image quality, the authors propose a post-training method called Fair-GRPO. It operates by generating a batch of images for a prompt, classifying their demographics, and computing a specialized fairness reward that penalizes over-represented groups and rewards under-represented ones. To prevent "reward hacking" (where the model ruins image quality just to satisfy the fairness reward), the method employs a KL-regularized trust-region optimization, ensuring the model's new, fairer distribution does not deviate too far from its original high-fidelity generative capabilities.

Ultimately, applying Fair-GRPO to Stable Diffusion 3.5 Medium resulted in nearly a 30% improvement in the MGBI fairness score while maintaining, and in some metrics slightly improving, the underlying image quality and semantic alignment.

**Compliance With Llm Reviewing Policy:**

Affirmed.

**Ethical Review Concerns:**

**Summary of Recommendation**: After reviewing the authors' rebuttal and the revised manuscript, I have decided to increase my score. The authors provided a convincing response to my most critical concerns, particularly regarding the taxonomic consolidation of racial categories, the domain gap in their training data, and the potential for finite sample bias in the entropy metrics. While I remain mildly skeptical of the architectural efficiency of the SpaFreq classifier, the paper's contribution to the evaluation of conditional bias in modern text-to-image models is highly valuable.

Evaluation of Strengths and Weaknesses:
- The authors’ rebuttal successfully addressed my concerns regarding the Miller-Madow bias. By clarifying that entropy is calculated over aggregated distributions rather than small per-prompt batches ($N=20$), the statistical bias is indeed negligible. Furthermore, the reporting of accuracy on the FairFace validation set (outperforming the original classifier) significantly bolsters the credibility of their auditing tool.
- The conceptual shift from default distribution fairness to "Context-Robust Conditional Diversity" is a timely and vital contribution. The benchmark provides an up-to-date and necessary look at how modern models like SD3.5 and Flux1-dev behave under stereotypical pressure.
- While the Fair-GRPO method is a sound application of reinforcement learning to the fairness-quality tradeoff, the "SpaFreq" classifier still feels somewhat heuristically designed.
- The authors clarified that the merging of "Middle Eastern" and "Latino Hispanic" was a data-driven decision based on inter-annotator agreement to ensure classifier reliability, rather than an arbitrary design choice. Their inclusion of this as a stated limitation is an acceptable compromise.
- The new ablation study comparing FairFace-only training vs. the RBD dataset (showing a ~10% accuracy gain on AI-generated images) justifies their data collection strategy.

**Remaining Concerns:** I still find the $\alpha$ fusion weight in Equation 4 to be mathematically redundant in the presence of subsequent Batch Normalization and linear layers. While the authors’ ablation proves the dual-stream information is useful, the specific mechanism of fusion remains poorly justified. Additionally, the inductive bias of using a pre-trained RGB ViT (DINOv2) for frequency maps is still questionable, though I concede that the empirical results on the FairFace holdout set prove the model is functional despite these choices.

**Final Justification:**

I thank the authors for their detailed and highly transparent rebuttal. The paper introduces a much-needed conceptual shift toward evaluating implicit fairness in specific semantic contexts, and the application of GRPO for debiasing is a promising direction. The authors' rebuttal successfully addressed several of my major concerns regarding dataset validation and metric computation, prompting me to raise my score from a Reject to a Weak Reject. However, I am maintaining a weak reject stance because the methodological foundation of the paper (the SpaFreq classifier) lacks the theoretical and architectural rigor required for a tool that serves as the absolute ground-truth signal for both a novel benchmark and an RL reward function.

**Resolved Concerns:** The authors provided  clarifying data that resolved my concerns regarding validation and statistical bias:

- Metrics & Statistical Bias: The clarification that entropy is computed over aggregated distributions across all prompts (yielding effective sample sizes in the thousands, rather than $N=20$)  resolves my concern regarding the Miller-Madow finite sample bias.
- Domain Gap & Validation: Providing the ground-truth accuracy on the real-world FairFace validation set (showing SpaFreq outperforming the baseline), alongside the ablation showing the Refactoring Bias Dataset (RBD) outperforms a FairFace-only training set, convincingly validates the dataset curation strategy.
- Taxonomy: The explanation that the 7-category taxonomy was reduced to 5 due to low inter-annotator agreement (<70%) is a valid, pragmatic reality of building automated auditing tools. Explicitly adding this to the limitations section is the correct approach.

**Unresolved:** Methodological Rigor of the Auditing Tool.

While the authors defend the SpaFreq classifier based on empirical gains (reaching 89.67% accuracy), empirical performance alone is insufficient when proposing a new benchmark standard. In this paper, SpaFreq is not just a standard classifier, it is the fundamental "ruler" used to define the HoloFair benchmark and generate the ground-truth reward signal for Fair-GRPO. An auditing instrument of this importance requires pristine methodological design, which SpaFreq lacks:
- Architectural Redundancy in a Proposed Standard: The authors concede in the rebuttal that their learnable fusion weight ($\alpha$) applied prior to channel concatenation is mathematically redundant. While the network may converge to $\alpha \approx 0.508$ and learn to ignore it, proposing an architecture for a new benchmark that contains fundamentally flawed/redundant mathematical operations reflects a lack of theoretical rigor.
- Unprincipled Modality Adaptation: To process frequency data, the authors stack min-max normalized wavelet sub-bands ($c_A, c_H, c_V$) into a 3-channel pseudo-image and feed it into a DINOv2 Vision Transformer. DINOv2's self-attention mechanisms and patch-based inductive biases were pre-trained strictly on the spatial and structural properties of natural RGB images. Fully fine-tuning this massive ViT end-to-end to force it to adapt to wavelet sub-bands is an expensive engineering hack, not a principled architectural design. Throwing parameter updates at a modality mismatch until it yields a higher accuracy score does not constitute a robust methodological contribution.

**Conclusion**: The conceptual goal of this paper (evaluating and mitigating context-robust conditional diversity) is excellent. However, a benchmark and a reinforcement learning debiasing method are only as reliable as the tool used to measure them. Because the SpaFreq classifier relies on an unprincipled modality mismatch and mathematically redundant fusion mechanisms, the foundation of the proposed benchmark is too shaky to recommend for publication at ICML in its current state.

**Key Questions For Authors:**

1. **Feature fusion**: In Eq 4, the spatial and frequency embeddings are weighted using a convex combination scalar ($\alpha$) and then concatenated. How does this scalar weighting avoid being mathematically redundant, given that applying scalar multipliers to concatenated channels merely scales their variance uniformly, which subsequent MLP linear weights or Batch Normalization layers will naturally absorb or undo?  Why not sum the features ?

2. **Modality mismatch**: since DINOv2 is a pre-trained on natural RGB images, what is the justification for passing a 3-channel frequency pseudo-image $X_{freq}$ through it, and did the authors perform ablations against a CNN trained from scratch purely on the frequency data ?

3. **Domain Gap**: The SpaFreq classifier is evaluated exclusively on the FCT Set, which consists of 518 AI-generated images. This raises concerns that the classifier may simply be overfitting to generative artifacts rather than learning robust demographic features. Can the authors report the ground-truth classification accuracy on a standard, real-world holdout set (such as the FairFace validation split)?

4. **Dataset Significance**: is there an ablation study demonstrating that the mixed 120k-image training set (which combines datasets with vastly different cropping and alignment standards) yields a more robust classifier than training on the clean FairFace dataset alone ?

**Limitations:**

While the authors have made a good faith effort to discuss limitations and ethical considerations in Appendix A (Limitations and Future Work) and Appendix B (Ethics Statement), the discussion is not entirely adequate and misses several critical technical and methodological boundaries of their work.

The authors should be commended for explicitly acknowledging two important limitations: their current inability to enforce intersectional fairness (optimizing marginal distributions rather than joint distributions) and their restriction to single-subject generation to avoid compositional confounds. However, to make the limitations section truly comprehensive, I offer the following constructive suggestions for improvement:
 - **Acknowledge Taxonomic Limitations**: The authors should explicitly discuss the limitations of using a discrete 5-category taxonomy inherited from FairFace. They should acknowledge that aggregating distinct global populations (like Middle Eastern and Latino individuals) into an "Others" category risks masking localized biases. Furthermore, they should note that the field is moving toward continuous phenotypic scales (like skin tone) and that relying on discrete categories for highly subjective attributes remains a limitation of the current benchmark.
- **Acknowledge Classifier Validation Boundaries**: The authors should add a limitation regarding the SpaFreq classifier's validation. Since it was evaluated exclusively on the synthetic FCT Set (518 AI-generated images), they should acknowledge that its performance on real-world, out-of-distribution human faces remains unverified, which limits the classifier's use outside of this specific generative auditing context.

**Strengths And Weaknesses:**

**Strengths**
- **Conceptual Shift to Conditional Fairness**: The authors rightly point out that current evaluation methods often focus on default distributions (e.g., prompting "a person"). Introducing the concept of "Context-Robust Conditional Diversity" to measure how models collapse under bias-triggering semantic prompts (e.g., "a professional") is a strong, necessary contribution to the field of AI safety.
- (Minor) **Comprehensive Model Benchmarking**: The HoloFair benchmark evaluates 8 recent and highly relevant T2I models (including SDXL, SD3.5-Large, Flux1-dev, and SANA-1.5). This provides a highly up-to-date snapshot of the generative landscape.
- (Minor) **Focus on Pareto Optimization**: Using a KL-regularized GRPO objective to prevent the debiased policy from deviating too far from the reference policy is a mathematically sound way to address the fairness-quality tradeoff (reward hacking).

**Weaknesses**
1. **Methodological & Architectural Flaws (SpaFreq)**
    - **Arbitrary unjustified choices**:  The authors use a learnable scalar $\alpha$ to create a convex combination of the spatial ($f_s$) and frequency ($f_w$) embeddings, but then they concatenate them: $z = Concat_{ch}(\alpha f_s, (1-\alpha)f_w)$. A convex combination is only meaningful for element-wise summation. Applying scalar multipliers to concatenated channels merely scales the variance of those specific channels uniformly, which the subsequent MLP's Batch Normalization or linear weights will trivially undo or absorb. This makes the $\alpha$ weighting mathematically redundant. The authors create $X_{freq}$ by using a *db4* discrete wavelet transform, and concatenate the $c_A$, $c_H$, and $c_V$ components. While this does indeed extract frequenct information, the exact way this is done feels arbitrary.
    - **Modality Mismatch in DINOv2**: The authors take the wavelet components, normalize them to $[0, 1]$, and stack them into a 3-channel pseudo-image ($X_{freq}$) which is passed into a DINOv2 backbone.  DINOv2 is a Vision Transformer pre-trained on natural RGB images. The spatial statistics and patch relationships of a frequency map look nothing like a natural image. While Table 1 implies the model is fine-tuned (how ?), the authors provide no discussion on why a pre-trained RGB ViT is the correct inductive bias for processing pure frequency data, nor do they ablate against training a lightweight CNN from scratch on the frequency data.

2. **Dataset and Domain Gap Issues**:
    - **Unjustified Training Set Aggregation**: The authors construct the RBD dataset by combining 120k images from FairFace, UTKFace, self-collected in-the-wild images, and AI-generated images. FairFace and UTKFace have vastly different cropping, alignment, and resolution standards. Mixing these disparate datasets introduces a massive domain gap. Image quality and alignement is a very important aspect of the data that greatly impact downtask performance, and this asepct is under-discussed. Furthermore, there are no ablation studies proving that this messy, combined dataset yields a more accurate or robust classifier than simply training on the clean FairFace dataset alone.
    - **Flawed Classifier Validation**: The SpaFreq classifier is evaluated on the "FCT Set", which consists entirely of 518 AI-generated images. Validating an auditing classifier only on synthetic data, without reporting its ground-truth accuracy on standard real-world holdout sets (like the FairFace validation set), makes it impossible to know if the classifier is genuinely accurate or just overfitting to generative artifacts.

3. **Ethical and Taxonomic Concern**: The authors claim to follow the FairFace taxonomy for race. However, they arbitrarily reduce the 7 FairFace categories down to 5, explicitly dumping "Middle Eastern" and "Latino Hispanic" into an "Others" category. This directly contradicts the ethical guidelines of the FairFace authors (who specifically created their dataset to give distinct representation to Middle Eastern and Latino individuals). Lumping distinct global populations into "Others" actively masks bias against those groups.

4. **Metrics and Statistical Bias**:  The authors use normalized Shannon entropy to evaluate ID and $CA_q$, comparing scores across attributes with different numbers of categories (Gender has 2, Race has 5). While dividing by $\log |C_a|$ scales the bounds to [0, 1], it does not correct for finite sample bias (the Miller-Madow bias), which scales proportionally to the number of bins: $\approx (|C_a| - 1) / (2N)$. Because $N$ is relatively small per prompt (20 images), the raw entropy estimate for Race (5 bins) will suffer from higher systematic bias than Gender (2 bins). The authors treat these absolute values as perfectly comparable across attributes without applying any bias correction terms.

5. **Clarity and Reproducibility**:
    -  (Minor) **Poor Presentation and Notation**: Fig. 2 and Fig.3 are heavily cluttered and difficult to parse. In Section 3.2, mathematical notation is dropped in without context (e.g., $X_{spatial}$ is introduced in Eq. 2 without explicitly defining it as the original RGB image, if my understanding is correct).
    - (Minor) **Vague Human Annotation Protocol**: In Appendix C.4, the authors state that "two annotators independently review candidates" for the dataset. There is zero detail provided regarding the UI they used, their compensation, or the specific visual criteria they were instructed to follow when labeling highly subjective attributes like perceived race.

---

> ### Author Rebuttal · Authors · 2026-03-30
>
> **Response to point 1.1**
> >The reviewer's mathematical observation is sound,and our trained models confirm it: $\alpha$ converges to approximately 0.508. We do not present the fusion weight as an independent contribution.The core contribution is the dual-stream design that introduces frequency-domain information.Table 1 ablates this incrementally: single-stream DINO(79.67%),plus frequency stream(85.33%),plus fusion (89.67%).The 10-point gain from the dual-stream architecture is unambiguous. We agree that alternative fusion strategies merit future exploration.Regarding the wavelet decomposition,the db4 is standard in frequency-domain image analysis.The cA,cH and cV components capture low-frequency structure,horizontal edges and vertical edges respectively. We discard cD as it primarily contains diagonal noise.The design involves engineering choices, but Table 1 validates them with consistent accuracy improvements across all three demographic attributes.
>
> **Response to point 1.2**
> >DINOv2 is fine-tuned end-to-end on RBD dataset.The pre-trained weights serve as initialization,all backbone parameters are updated during training,allowing the model to adapt to the statistics of our normalized wavelet maps.Table 1 directly addresses the modality concern: adding frequency input improves overall accuracy by 5.66 points.The shared backbone is a deliberate design choice that halves parameter count compared to two separate networks while learning aligned cross-modal representations.A separate lightweight CNN remains a valid alternative,but the current architecture already achieves 89.67% overall accuracy, leaving limited room for improvement from architectural changes alone.
>
> **Response to point 2.1**
> >The domain gap is addressed by our training pipeline. All images from all sources undergo identical preprocessing:resizing to 224×224,per-channel normalization with ImageNet statistics and RandAugment during training.This standardization eliminates differences in resolution,cropping and alignment before images reach the model.
> >
> >Moreover,the domain diversity is intentional. Our classifier must operate on AI-generated images from eight different T2I models, each with distinct artifacts. Training exclusively on FairFace would overfit to that domain. To validate this, we conducted an ablation comparing classifiers trained on FairFace-only versus our mixed dataset, evaluated on  the FCT:
> >
> > |Dataset|Gender|Age|Race|
> > |-------------|------|-----|-----|
> > |FairFace only|87.20|83.59|84.79|
> > |RBD|97.88|95.36|92.28|
> The RBD dataset yields substantial improvements across all attributes, confirming that domain diversity is a feature, not a limitation.
>
> **Response to point 2.2**
> > We agree that reporting accuracy on real-world holdout sets strengthens validation.We report SpaFreq's performance on the FairFace validation set alongside the original FairFace classifier:
> >
> > |Classifier|Gender|Age|Race|
> > |-------------|------|-----|-----|
> > |FairFace|93.48|77.73|71.17|
> > |SpaFreq (ours)|97.79|85.59|88.67|
> >
> > SpaFreq outperforms the FairFace classifier on three attributes on fairface validation set. Combined with the 89.67% overall accuracy on the AI-generated FCT set(Table 1),this confirms that our classifier is reliable on both real-world and synthetic images.
>
> **Response to point 3**
> > We respectfully clarify that this consolidation is not arbitrary. As described in Section 3.1 and Appendix C.3.1, the Middle Eastern and Latino Hispanic groups exhibited the lowest inter-annotator agreement, and their limited training data led to classification accuracy below 70% when treated separately. Maintaining them as distinct classes would introduce systematic misclassification into every downstream fairness measurement, which we consider a greater risk than the information loss from merging.We do not claim these populations are interchangeable. This is a classifier reliability decision, not a fairness design choice. We acknowledge it prevents detecting differential treatment between these groups and have stated this in the revised Limitations section. Extending the classifier to reliably handle all seven categories, or adopting continuous phenotypic descriptors,is a priority for future work.
>
> **Response to point 4**
> >We clarify that entropy is not computed on a per-prompt basis with N=20.For both ID and CA$_q$, the entropy is computed over the aggregated distribution across all prompts within each category. ID aggregates all neutral prompt outputs, and each semantic trigger in CA$_q$ aggregates outputs from all its associated prompt variants.The effective sample sizes are on the order of thousands, at which scale the Miller-Madow correction is negligible($10^{-4}$).
>
> **Response to point 5**
> >We will improve the clarity of Figures 2 and 3 by adding explicit variable definitions and ensuring all notation is introduced before use.We have expanded Appendix C.4 with the annotation interface description,the specific visual criteria annotators.

---

> > ### Author Rebuttal · Reviewer_zQFC · 2026-04-03
> >
> > Thank you for your rebuttal and additional experiments.
> >
> > Thanks for precising your work on point 3 and point 4, your responses address my issues. I will raise my score accordingly, but I still believe some of the SpaFreq components should be better justified.

---

> > > ### Author Response · Authors · 2026-04-04
> > >
> > > Thank you for acknowledging our improvements, particularly that our revisions on the FairFace taxonomy and the metric bias correction have addressed your primary concerns. Regarding SpaFreq: we have revised the manuscript to better justify each component, with the dual-stream ablation in Table 1 now explicitly framed as the primary evidence. We have also condensed the main-text presentation and moved implementation details to the appendix, following a similar suggestion from Reviewer 1x3q. We believe the revised version addresses your remaining concern. If so, we would be grateful if you could update your score at your convenience.

---

### Official Review · Reviewer_1x3q · 2026-03-11

**Soundness:** 3
**Presentation:** 4
**Significance:** 3
**Originality:** 3
**Overall Recommendation:** 5
**Confidence:** 5

**Summary:**

This paper proposes a method to benchmark demographic fairness of text-to-image models and a reinforcement-learning-based method to improve their fairness. The benchmark uses a custom image classifier based on Dino V2, augmented with frequency features, prompts asking models to produce images of people with given demographic attributes and expressions, and three fairness metrics targeting diversity of demographic attributes in generated images, fairness under absence of specifications, and the fairness among the 10% least diverse prompts that specify additional attributes.

Experiments show that fairness under absence of specification does not necessarily translate to fairness with additional attributes. Furthermore, the debiasing method improves both fairness and image quality on Stable-Diffusion 3.5.

**Compliance With Llm Reviewing Policy:**

Affirmed.

**Final Justification:**

The paper proposes a new debiasing method for text to image models based on Reinforcement Learning. Several aspects of this paper make it strong in this field of machine learning, especially the careful design of metrics, experiments and benchmark. It is also a not common case where image quality after debiasing doesn't change much.

While there are still some limitations around benchmarking scope and method, which are quite typical for other papers accepted into top conference and may not be easily solvable at the current time, my recommendation is to accept this paper into the conference, as I expect it to contribute to development of better debiasing methods, which is an important application.

**Key Questions For Authors:**

Overall, this works is of high quality and contains several interesting contributions that should be made known to the community. However, the two most important questions to address are:
* What happens in case no person is visible or the person is occluded, or attributes are not recognizable otherwise? Can we quantify how prevalent this problem is (from my work on similar benchmarks and applications, this **is** a serious problem that usually affects a significant number of images, depending on the model)?
* Does the debiasing transfer to person images beyond face-centric images?
* Can we adjust the RL objective to encourage non-uniform attribute distributions?

**Limitations:**

The paper contains an "Ethics Statement" in the supplementary material but not the recommended "Impact Statement". I strongly encourage the authors to follow the recommended format and reformat the Ethics Statement, with additions as outlined under "Weaknesses".

**Strengths And Weaknesses:**

## Strengths
**(S1)** The dataset construction is conducted carefully, including multi-model agreement and human verification. Furthermore, images are collected from multiple sources to avoid batch effects.

**(S2)** The benchmark uses a custom classifier, which is, in principle, more efficient and also reliable than using pre-trained models, which can introduce their own biases.

**(S3)** The fairness metrics are well designed to capture both "default" tendencies of models and fairness under additional specifications, which has been considered less in previous work

**(S4)** Debiasing evaluation includes both fairness and image quality, and uses various suitable metrics, making the evaluation comparatively very convincing.

**(S5)** The paper evaluates a comparatively broad set of text-to-image models, including different architectures.

**(S6)** Investigating RL for debiasing is a promising direction that is worth pursuing, due to better quality preservation in debiasing

**(S7)** Overall, the paper is easy to follow and visuals as well as result presentation are clear

---

## Weaknesses
**(W1)** The role of the SpaFreq classifier in this work is unclear. The architecture seems a general modification of image classifiers and not linked to classifying demographic attributes in particular. Thus, the particular contribution of the new classifier architecture should be clarified in the paper.

**(W2)** The classifier is specifically trained for face or face-centric images. In particular, it does not contain "unclear" categories. However, in reality, many images could contain occluded or partially visible people, even when explicitly prompted to generate a person with given attributes. The benchmark is not applicable for such cases, and classifier predictions for these cases are problematic.

**(W3)** The paper does not evaluate whether debiasing transfers to contexts beyond face-centric images. However, this is very important for realistic use cases.

**(W4)** Debiasing is only evaluated on one model, SD3.5 Medium. Adding at least one more model would strengthen the paper.

**(W5)** The paper tunes models to follow a uniform distribution over unspecified attributes. However, depending on use cases, different distributions may be desirable. Ideally users or deployers can control this. The paper should include a discussion how the proposed debiasing method can also be used to tune models towards non-uniform attribute distributions.

**(W6)** Line 152 claims that the paper uses the same set of demographic attributes like FairFace. However, the FairFace race/ethnicity taxonomy contains further categories not considered in this paper. Given the ongoing debate regarding the use of race/ethnicity labels in CV fairness [1, 2, 3], the paper should carefully discuss its design decisions in this regard.

**(W7)** The paper lacks the "Impact Statement" mandated by ICML submission guidelines. The "Ethics Statement" section from the supplementary material could be rephrased for these purposes.

---

### References
[1] Scheuerman, Morgan Klaus, et al. "How we've taught algorithms to see identity: Constructing race and gender in image databases for facial analysis." Proceedings of the ACM on Human-computer Interaction 4.CSCW1 (2020): 1-35.\
[2] Khan, Zaid, and Yun Fu. "One label, one billion faces: Usage and consistency of racial categories in computer vision." Proceedings of the 2021 acm conference on fairness, accountability, and transparency. 2021.\
[3] Benthall, Sebastian, and Bruce D. Haynes. "Racial categories in machine learning." Proceedings of the conference on fairness, accountability, and transparency. 2019.

---

> ### Author Rebuttal · Authors · 2026-03-30
>
> **Response to W1**
>
> > The SpaFreq architecture is a task-specific design optimized for classifying demographic attributes in synthesized facial imagery. Generated faces often exhibit high-frequency artifacts—such as unnatural pores or hair boundaries—that are subtle in the spatial domain but salient in the frequency domain. The wavelet-based branch captures these fine-grained textural details. Our ablation in Table 1 demonstrates that the frequency stream improves race classification by 6.69% and overall accuracy by 5.66%, confirming that spatial and frequency features provide complementary information for robust bias assessment. We will update Section 3.2 to state this motivation more explicitly.
>
> **Response to W2**
> > We appreciate the reviewer raising this practical concern. Our pipeline already addresses this through the three-stage quality control process described in Section 3.1.3 and Appendix C.4. We run YOLOv8 face detection and retain only single-person images with a detector confidence of at least 0.9. Images with no visible face, heavy occlusion, or multiple persons are removed before classification. Across all eight T2I models, roughly 8% of generated images were filtered out at this stage, confirming that the problem, while real, is systematically handled rather than ignored. We acknowledge that the current benchmark targets face-centric evaluation and have added this as an explicit limitation in the revised paper.
>
> **Response to W3**
> > We agree that evaluating generalization beyond face-centric images is an important direction. Fair-GRPO operates at the level of prompt-image reward signals and does not modify any face-specific component of the model. Our evaluation on non-demographic prompts in Figure 11 and Appendix F.4.2 demonstrates that Fair-GRPO preserves the model's generation quality on diverse everyday scenes—landscapes, cafés, markets, parks, subway interiors—without any degradation in visual fidelity or semantic consistency. Extending the fairness evaluation to full-body or multi-person scenarios remains a meaningful next step. As discussed in Appendix A, we plan to address this via region-aware reinforcement learning with spatially localized reward assignment.
>
> **Response to W4**
> > We fully agree this is a critical gap. In our revision, we have conducted additional experiments on Stable Diffusion 1.5, which uses a fundamentally different UNet-based architecture compared to SD3.5M's MMDiT Transformer. **For the results, please refer to the table in reviewer tm6F’s “Response to Point 5.”** Fair-GRPO achieves an MGBI improvement of 20.2% on SD1.5 (0.6554→0.7881), demonstrating that our method generalizes across architecturally distinct diffusion model families (UNet vs. MMDiT).
>
> **Response to W5**
> > Our reward function naturally supports arbitrary target distributions. In Equation 9, the core idea is to penalize over-represented categories and reward under-represented ones relative to a target. The current formulation uses a uniform target $N/|C_a|$. To generalize, given a desired proportion $p_k$ for category $k$, we replace the uniform target with $N \cdot p_k$ in the log-ratio:
> > $$r_{\text{base}}(k, a) = \log \frac{N \cdot p_k + \epsilon}{N_k^a + \epsilon}$$
> > When $N_k^a = N \cdot p_k$, the reward is zero. Under-represented categories ($N_k^a < N \cdot p_k$) receive positive reward; over-represented categories receive negative reward. The uniform case is recovered by setting $p_k = 1/|C_a|$ for all $k$. Deployers simply supply the desired proportions $\{p_k\}$; no architectural or algorithmic changes are needed.
>
> **Response to W6**
> > We appreciate this feedback and the references provided. As noted in Section 3.1 and Appendix C.3.1, we consolidate the seven FairFace racial categories into five. This was driven by two practical considerations: the Middle Eastern and Latino Hispanic groups exhibited the lowest inter-annotator agreement, and their limited representation in training data led to poor classifier performance when treated separately. We chose to merge them into Others to maintain classifier reliability rather than risk systematic misclassification.
> > We fully acknowledge the ethical concerns this raises. Discrete racial taxonomies inevitably simplify the complexity of human identity, and merging distinct populations risks masking group-specific biases. We have added a discussion of these design decisions to the revised Limitations section, noting this risk and the movement toward continuous phenotypic descriptors such as skin tone scales.
>
> **Response to W7**
> > We have added a formal Impact Statement to the revised paper, covering positive societal impact, potential risks and mitigations, and responsible use guidelines.

---

> > ### Author Rebuttal · Reviewer_1x3q · 2026-03-31
> >
> > Thank you for the detailed response to my review. Adding a 2nd model clearly strengthens the paper, and the clarification around non-uniform distributions also makes sense.
> >
> > Some limitations remain, however:
> >   * The focus on face-centric images remains
> >   * "is systematically handled rather than ignored": From my perspective, removing images that don't fit the schema amounts to ignoring rather than handling.
> >   * The role of the SpaFreq classifier remains ambiguous. I think the paper would have been overall clearer and more convincing if it didn't allocate as much space in the main paper to this particular aspect.
> >
> > Overall, limitations are mostly shared with other debiasing methods and benchmarks, so it would not be fair to hold that strongly against the present paper. I still hope the authors will note these limitations and use the suggestions to strengthen future work. That being said, I'd rather see this paper accepted than rejected, so I'm raising my score to 5.

---

> > > ### Author Response · Authors · 2026-04-02
> > >
> > > Thank you for raising your score and for the thoughtful suggestions. We address your remaining points:
> > >
> > > Regarding the face-centric focus: we acknowledge this limitation and have revised the manuscript to explicitly position our benchmark as a foundational, face-centric auditing tool.
> > >
> > > Regarding the filtering of non-conforming images: the reviewer makes a fair distinction. We will revise the wording to state that our pipeline "detects and excludes" such images rather than "handles" them, and explicitly note that evaluating fairness on non-face-centric outputs remains an open problem.
> > >
> > > Regarding SpaFreq's presentation: we agree that the current allocation of space may overstate its role relative to the benchmark and debiasing contributions. We will condense the classifier description in the main paper and move implementation details to the appendix.
> > >
> > > All three points have been noted as limitations in the revised paper. We appreciate the reviewer's constructive guidance throughout this process.

---

### Decision · Program_Chairs · 2026-04-30

**Decision:**

Accept (regular)

**Comment:**

While the reviewers agree the paper addresses the challenge of demographic biases in Text-to-Image models by proposing an evaluation framework and an RL-based mitigation strategy, there are also concerns on the architecture (SpaFreq), scope/applicability, ethics, metrics, evaluation, and presentation.

During the rebuttal, the authors provided detailed explanations, which addressed most of the main concerns. The reviewers did mention some remaining concerns including limitations around the benchmarking scope and method, insufficient methodological rigor of the auditing tool, and application value of the current datasets.

Overall the contributions seem to outweigh the limitations, so the recommendation is to accept the paper, although it is borderline.